

# Observing electric field and neutral wind with EISCAT 3D

Johann Stamm[1], Juha Vierinen[1], and Björn Gustavsson[1]

[1]Institute for physics and technology, University of Tromsø, Tromsø, Norway

**Correspondence:** Johann Stamm (johann.i.stamm@uit.no)

**Abstract.** Measurements of height dependent electric field ($E$) and neutral wind ($u$) are important governing parameters of the Earth's upper atmosphere, which can be used to study e.g., how auroral currents close, or how energy flows between the ionized and neutral constituents. The new EISCAT 3D (E3D) incoherent scatter radar will be able to measure a three-dimensional ion velocity vector ($v$) at each measurement point, which will allow less stringent prior assumptions about $E$ and $u$ to be made
when estimating them from radar measurements. This study investigates the feasibility of estimating the three-dimensional electric field and neutral wind vectors along a magnetic field-aligned altitude profile from E3D measurements, using the ion momentum equation and Maxwell's equations. The uncertainty of ion drift measurements is estimated for a time and height resolution of 5 s and 2 km. With the most favourable ionospheric conditions, the ion wind at E region peak can be measured with an accuracy of less than 1 m/s. In the worst case, during a geomagnetically quiet night, the uncertainty increases by a
factor of around ten. The uncertainty of neutral wind and electric field estimates is found to be strongly dependent on the prior constraints imposed on them. In the lower E region, neutral wind estimates have a lower standard deviation than 10 m/s in the most favourable conditions. In such conditions, also the F region electric field can be estimated with uncertainty of about 1 mV/m. Simulated measurements of $v$ are used to demonstrate the ability to resolve the field-aligned profile of $E$ and $u$. However, they can only be determined well at the heights where they significantly influence the ion drift, that is above
125 km for $E$ and below 115 km for $u$. At the other heights, the results are strongly dependent on the the prior assumptions of smoothness.

## 1   Introduction

One of the main parameters that incoherent scatter radars (ISR) measure is ion drift velocity $v$. This can be related primarily to electric field $E$ and neutral wind $u$, making it possible to use ISR measurements for estimating these parameters, which
are of interest when e.g., studying the electrodynamics of the aurorae borealis (e.g., Takahashi et al., 2019), determining how auroral currents close within the ionosphere, and studying how energy is transferred between the ionosphere and the neutral atmosphere (e.g., Aikio and Selkälä, 2009; Cai et al., 2016).

The method for simultaneously estimating $E$ and $u$ for the auroral ionosphere using an incoherent scatter radar was first described by Brekke et al. (1973). Since then, this technique has been used and improved (see Nygrén et al., 2011, and
references therein). The velocity of both the ion and neutral wind is related to other ionospheric parameters, such as ion-neutral collision frequency and electric field through the momentum equation of the ions. Some parameters can be measured while





others need to be taken from models. The terms with pressure gradients and gravitation are commonly neglected. The electrical field can be deduced from measurements higher up where ion-neutral collisions are neglible and used further down by assuming that electric field along the magnetic field line is constant (Brekke et al., 1994; Heinselman and Nicolls, 2008).

Currently ion-drifts are measured with a monostatic radar by pointing the transmit beam to three or more different directions and measuring the projection of the ion-velocity vector onto these directions. If the ion velocity then is assumed constant or slowly changing in the horizontal direction for all of the pointing directions, an estimate of the ion drift can be made (Heinselman and Nicolls, 2008; Nicolls et al., 2014b). If the observations are made with slowly moving dish-based radars, making the measurements in different pointing directions might take several minutes (Williams et al., 1984). With modern

phased array systems the scanning time can be reduced (Heinselman and Nicolls, 2008).

Multistatic radars can contribute to the measurement challenge by simultaneously measuring a common scattering volume from multiple different directions. Provided at least three linearly independent ion velocity measurements, the full vector can be determined. With dish-antennas, the ion velocity is obtained only from one intersection volume at a time (Williams et al., 1984; Risbeth and Williams, 1985). In order to obtain measurements along a line or over a volume of space, the antennas need to

be steered, which again takes time. In 2008, only the EISCAT UHF system could offer multistatic measurements (Heinselman and Nicolls, 2008). Later, the receiver antennas in Kiruna and Sodankylä were converted to receive the signals from EISCAT VHF instead (Kero, 2014; Mann et al., 2016). Earlier, there have also been other ISRs that have been multistatic (Williams et al., 1984).

One of the capabilities of the upcoming EISCAT3D (E3D) incoherent scatter radar is that it can simultaneously observe

scatter from ionospheric plasma from at least three different geographically separated receiver sites, each using multiple simultaneous receiver beams that intersect the transmit beam at multiple heights. This is made possible by the use of phased array technology (f.ex. Wirth, 2001), which allows fast beam steering and the receivers to form multiple simultaneous beams. A measurement of the ion velocity vector along the radar transmit beam is then possible without any other assumptions than homogeneity of ion velocity within the common scattering volumes where the transmit and receive beams intersect (McCrea

et al., 2015; Virtanen et al., 2014).

## 2   Ion wind

ISR measurements mainly provide four ionospheric parameters: Electron density $n_\mathrm{e}$, electron temperature $T_\mathrm{e}$, ion temperature $T_\mathrm{i}$, and the ion velocity component along the Bragg scattering vector $w_p$. The connection between $w_p$ and the Bragg scattering vector $\boldsymbol{k_p}$ for transmit-receive-pair $p$, and the ion velocity $\boldsymbol{v}$ is

$$w_p = \boldsymbol{k_p} \cdot \boldsymbol{v}/|\boldsymbol{k_p}| + \varepsilon_p, \tag{1}$$

where $\varepsilon_p$ is a random variable that models the velocity measurement errors.





The basis for finding the velocity of the uncharged wind and electric field is through the ion velocity. The measurements $w_p$ and the unknown velocity $\boldsymbol{v}$ can be set up as a linear inverse problem (Heinselman and Nicolls, 2008; Nygrén et al., 2011):

$$\boldsymbol{w} = \mathbf{K}\boldsymbol{v} + \boldsymbol{\varepsilon}, \tag{2}$$

where $\boldsymbol{w}^\top = [w_1, ... w_P]$ is a vector containing independent measurements of ion-line Doppler shift, $\mathbf{K}^\top = [\boldsymbol{k_1} ... \boldsymbol{k_P}]$ is the theory matrix, and $\boldsymbol{\varepsilon}$ is the noise vector. We assume that the noise is independent and identically normal distributed with zero mean and variance of $\sigma_w^2$ which means that we assume that the line-of-sight (LOS) ion velocity measured with different receivers have the same uncertainty.

With the first stage of E3D, there will be $P = 3$ measurements of the ion velocity. Since the theory matrix then is quadratic, it
will be possible to find $\boldsymbol{v}$ with low uncertainty and without restrictions if the measurements are sufficiently independent linearly (cf. Aster et al., 2013; Risbeth and Williams, 1985). The Bragg scattering vectors can be calculated from the preliminary positions of E3D as mentioned by Kero et al. (2019). We assume a target in the direction of the magnetic field at ionospheric range, extending outwards from the Skibotn transmitter site.

As it later will become an advantage to measure the ion velocity in magnetic field coordinates, we have to transform the
scattering vector matrix $\mathbf{K}$. The transformation matrix from geographic to local magnetic coordinates is

$$\mathbf{R}_{\text{geo}\rightarrow\text{gmag}} = \begin{bmatrix} \cos\delta & -\sin\delta & 0 \\ \sin I \sin\delta & \cos\delta \sin I & \cos I \\ -\cos I \sin\delta & -\cos I \cos\delta & \sin I \end{bmatrix} \tag{3}$$

where $\delta$ is the declination and $I$ is the dip angle of the magnetic field (Heinselman and Nicolls, 2008). The LOS velocities are then related to the ion velocity in local magnetic coordinates as follows,

$$\boldsymbol{w} = \mathbf{K}\mathbf{R}_{\text{geo}\rightarrow\text{gmag}}\boldsymbol{v} + \boldsymbol{\varepsilon}. \tag{4}$$

We find the ion velocity by solving the inverse problem by using the linear least squares method:

$$\hat{\boldsymbol{v}} = \left(\mathbf{K}^\top\mathbf{K}\right)^{-1}\mathbf{K}^\top\boldsymbol{w}. \tag{5}$$

The uncertainty of the ion velocity estimate is quantified using the following covariance matrix:

$$\boldsymbol{\Sigma}_v = \left(\mathbf{K}^\top\mathbf{K}\right)^{-1}\sigma_w^2. \tag{6}$$

We will use this uncertainty later when estimating electric field and neutral wind.

## 2.1 Uncertainty of ion wind velocity

Vallinkoski (1989) describes a method for finding the uncertainty of ISR parameter estimates. Our procedure is similar. Like the other ionospheric parameters, the ion velocity component along the Bragg scattering vector $w_p$ is estimated from the autocorrelation function (ACF) that is measured by the radar. The ACF $\boldsymbol{\rho} = [\rho(\tau_0), \rho(\tau_1), \ldots, \rho(\tau_{T-1})]^\top$ is described by the





theory for incoherent scatter (Kudeki and Milla, 2011). Here, $\tau$ is the time lag. The theory provides a non-linear relationship
between the parameters $\boldsymbol{\theta}$ and the ACF $\boldsymbol{\rho}$.

$$\boldsymbol{\rho} = \boldsymbol{f}(\boldsymbol{\theta}) + \boldsymbol{\varepsilon}, \tag{7}$$

The parameters $\boldsymbol{\theta}$ includes ionospheric plasma parameters and parameters specific to the radar experiment. In this relationship, there are also measurement errors, which are modelled with a random variable $\varepsilon$.

To simplify the uncertainty calculations, we linearize the relationship between ACF and the parameters measured with the
ISR. The first-order Taylor polynomial for the ACF around a parameter estimate $\boldsymbol{\theta}'$ is

$$\boldsymbol{\rho} = \boldsymbol{f}(\boldsymbol{\theta}') + \mathbf{J}(\boldsymbol{\theta} - \boldsymbol{\theta}') + \boldsymbol{\varepsilon}, \tag{8}$$

where

$$\mathbf{J} = \begin{bmatrix} \frac{\mathrm{d}\rho_0}{\mathrm{d}\theta_1} & \cdots & \frac{\mathrm{d}\rho_0}{\mathrm{d}\theta_n} \\ \vdots & \ddots & \vdots \\ \frac{\mathrm{d}\rho_{T-1}}{\mathrm{d}\theta_1} & \cdots & \frac{\mathrm{d}\rho_{T-1}}{\mathrm{d}\theta_n} \end{bmatrix}_{\boldsymbol{\theta} = \boldsymbol{\theta}'} \tag{9}$$

is the Jacobian of $\boldsymbol{f}$ evaluated at $\boldsymbol{\theta} = \boldsymbol{\theta}'$. We move the constant parameters over to the left side and get an inverse problem with
the solution

$$\hat{\boldsymbol{\theta}} = \left(\mathbf{J}^H \boldsymbol{\Sigma}_\rho^{-1} \mathbf{J}\right)^{-1} \mathbf{J}^H \boldsymbol{\Sigma}_\rho^{-1} \left(\boldsymbol{\rho} - \boldsymbol{f}(\boldsymbol{\theta}') + \mathbf{J}\boldsymbol{\theta}'\right), \tag{10}$$

where the superscript $^H$ denotes Hermitian transpose. The linearized covariance matrix quantifying the uncertainty of the estimate is

$$\boldsymbol{\Sigma}_{\hat{\theta}} = \left(\mathbf{J}^H \boldsymbol{\Sigma}_\rho^{-1} \mathbf{J}\right)^{-1}. \tag{11}$$

The uncertainty is dependent on how well the ACF is measured. This depends on the signal-to-noise ratio. The signal strength is dependent on the ionospheric plasma parameters as mentioned above, the radar equation, and the experiment design with pulse length, coding etc. The noise level is determined by the system noise temperature, which depends on the implementation of the receiver electronics and the sky noise temperature at the radar frequency.

To determine the ACF, we calculate the ISR spectrum as described by Kudeki and Milla (2011) and take its inverse Fourier
transform. We then multiply it with the signal strength which we take from the radar equation

$$P_{\mathrm{S}} = \frac{P_{\mathrm{t}} G_{\mathrm{t}} G_{\mathrm{r}} \lambda^2 \sin^2 \chi}{(4\pi)^3 R_{\mathrm{t}}^2 R_{\mathrm{r}}^2} \cdot \mathcal{V} n_{\mathrm{e}} \cdot \frac{(4\pi r_{\mathrm{e}}^2)}{1 + T_{\mathrm{e}}/T_{\mathrm{i}}}, \tag{12}$$

where $P_{\mathrm{t}}$ is transmit power, $G_{\mathrm{t}}$ is transmit gain, $G_{\mathrm{r}}$ is receive gain, $\lambda$ is the radar wavelength, $R_{\mathrm{t}}$ and $R_{\mathrm{r}}$ are the distance between the target and transmitter and receiver, $\chi$ is the polarization angle, and $r_{\mathrm{e}}$ is the classical electron radius. The scattering volume $\mathcal{V}$ is approximated as a spherical sector

$$\mathcal{V} = \frac{2\pi \Delta r}{3} \left(1 - \cos\frac{\varphi}{2}\right) \left(\frac{\Delta r^2}{4} + 3R_{\mathrm{t}}^2\right), \tag{13}$$





where $\Delta r$ is the resolution in range direction and $\varphi$ is the one-way half-power beamwidth of the radar. For bistatic cases with receiving in Karesuvanto or Kaiseniemi, we assume that the receiver sees the whole scattering volume such that we do not need to include possible losses because the radar beams do not overlap completely. The noise power $P_{\mathrm{N}}$ is obtained using the Nyquist-Johnson noise model as follows:

$$P_{\mathrm{N}} = k_{\mathrm{B}} T_{\mathrm{sys}} f_{\mathrm{B}} \tag{14}$$

where $k_{\mathrm{B}}$ is the Boltzmann constant, $T_{\mathrm{sys}}$ is the system noise temperature, and $f_{\mathrm{B}}$ is the bandwidth of the signal which is given by $f_{\mathrm{B}} = c/(2\Delta r)$ where $c$ is the speed of light.

The calculations require that measurements of the different lags of the ACF are uncorrelated. This means that the covariance matrix of the measurement errors $\boldsymbol{\Sigma}_\rho$ is diagonal. We can make this assumption if the different lags are measured using a coded long pulse with a low signal-to-noise ratio (Lehtinen and Häggström, 1987). A solution for cases with signal-to-noise ratio over 1 can be to shorten the baud length and so increase the resolution in range direction (Lehtinen and Damtie, 2013). This results in a weaker signal from every range, but provides more independent measurements that can be averaged to obtain the desired range resolution.

We can use this outline to calculate the uncertainty in $w_p$ for several representative cases. For the radar parameters of E3D, we use frequency $f$=233 MHz, one-way half-power beamwidth $2°$, both transmit and receive gain equal to 38 dB, transmit power 5 MW, and a noise temperature of 200 K. This takes into account the recent downscaling of the core array from 109 to about 35 subarrays for the first stage of E3D. We use a scattering angle of $90°$ even if it is not absolutely correct when receiving in Karesuvanto and Kaiseniemi.

In order to investigate the performance of the radar in different geophysical conditions, we have studied three different cases 1) day-time, 2) quiet night-time, and 3) night-time with auroral precipitation. Each of these cases have different ionospheric plasma parameter profiles consisting of $n_{\mathrm{e}}$, $T_{\mathrm{e}}$, $T_{\mathrm{i}}$ and $m_{\mathrm{i}}$. The key parameter that influences observability is $n_e$, as the signal-to-noise ratio is to first order proportional to this parameter. For the representative cases, we used the plasma-parameters for 2014-02-20 at three times: 14:00, 23:00, and 21:20 UT. The profiles are calculated by the IRI-2016 model (Bilitza et al., 2017), except for the aurora case at 21:20, where we used data from EISCAT UHF for electron density and the temperatures. We integrated the EISCAT data over 10 minutes in order to obtain plasma-parameters with small uncertainties. For calculating the magnetic field, we use the international geomagnetic reference field (see Thébault et al., 2015). The ionospheric parameter profiles for the three representative cases are shown in Fig. 1.

For the analysis, we assumed an experiment where the baud length is 15 μs, the pulse consists of 51 bauds, an interpulse period of 5 ms, and an integration time of 5 s. We use an analysis range resolution of 2250 m, corresponding to the baud length.

The uncertainty of the LOS ion velocities are shown in Fig. 2. According to the figure, the uncertainty at daytime and auroral nighttime are considerably lower than at nighttime without aurora. While the uncertainty varies from about 5 m/s at 100 km to 30 m/s at 140 km altitude in the non-aurora night case, for daytime and auroral nighttime conditions the uncertainty is smaller than 3 m/s. In general, the uncertainty is smaller where the signal-to-noise ratio is high. This occurs primarily at E region





**Figure 1.** Ionospheric parameter profiles we used to calculate the ion velocity errors.

heights, where the electron density is comparatively high. At F region heights the electron density is also high. However, this
is about twice as far as the E region, and the backscattered signal is therefore weaker.

It is worth noting that the test case is close to a solar maximum, which means that the electron density is comparatively high.
At solar minimum, the electron density in the ionosphere is in general lower (f.ex. Brekke, 2013), and the uncertainty in ion
velocity will be higher. One can compensate for this by integrating the LOS ion velocity over a larger number of range gates,
leading to a reduced range resolution. For example, Nygrén et al. (2011) used 10 km range resolution at E region heights in an
experiment with EISCAT UHF.

Using Eq. (6), we obtain the uncertainties of the ion velocity components, which are plotted in Fig. 2b-d. The uncertainty
in magnetic field-aligned component is very similar to the LOS uncertainties. This is expected because all LOS do not differ

**Figure 2.** $1\sigma$-uncertainty in line-of-sight velocity (a), and the hence following three components of the ion velocity vector in magnetic field coordinates: Perpendicular east (b), perpendicular north (c), and field-aligned direction (d)





much from the magnetic field line direction. Therefore, the uncertainty of the ion velocity components perpendicular to the magnetic field line is a factor 3-5 times higher. At the highest altitudes, the scattering vectors are even more similar, which

leads to an increased uncertainty.

## 3   Neutral wind and electric field

The velocity of ion and neutral wind are coupled through collisions as described by the ion momentum equation. This can be found by taking the first moment of the Boltzmann equation (f.ex Inan and Gołkowski, 2011). We assume that we can treat the ions as a single fluid. The momentum equation is

$$
n_i m_i \left[ \frac{d\boldsymbol{v}}{dt} + (\boldsymbol{v} \cdot \nabla)\boldsymbol{v} \right] = -\nabla \mathbf{P}_i + n_i m_i \boldsymbol{g} +
$$
$$
q_i n_i (\boldsymbol{E} + \boldsymbol{v} \times \boldsymbol{B}) -
$$
$$
\sum_k n_i m_i \nu_{ik} (\boldsymbol{v} - \boldsymbol{v_k}), \tag{15}
$$

where $n_i$ is the number density of ions, $m_i$ is the ion mass, $\mathbf{P}_i$ is the ion pressure tensor, $\boldsymbol{g}$ is the gravitational acceleration, $q_i$ is the ion charge, $\boldsymbol{E}$ is the electrical field, $\boldsymbol{B}$ is the background magnetic field, $\nu_{ik}$ is the momentum transfer collision frequency

between ions and particle species $k$, and $\boldsymbol{v}_k$ is the velocity of particle species $k$. We assume that spatial variations of the ion velocity are small such that we can neglect the term $(\boldsymbol{v} \cdot \nabla)\boldsymbol{v}$. Further we assume that the pressure is isotropic so we can write the pressure tensor as a scalar $p_i$. Only collisions between ions and neutrals are of importance to change the ion velocity (Brekke, 2013), other collision terms can be neglected. If the ions obey the ideal gas law, the ion pressure $p_i$ can be written as $p_i = n_i k_B T_i$. Additionally, we neglect local temperature variations such that $\nabla T_i = 0$. Finally, as in previous work, we also

neglect the contribution from pressure gradients and gravity. With all these assumptions, Eq. (15) can be rewritten as

$$
n_i m_i \frac{d\boldsymbol{v}}{dt} = q_i n_i (\boldsymbol{E} + \boldsymbol{v} \times \boldsymbol{B}) - n_i m_i \nu_{in} (\boldsymbol{v} - \boldsymbol{u}), \tag{16}
$$

where $\boldsymbol{u}$ is the neutral wind velocity.

For steady state conditions ($\frac{d\boldsymbol{v}}{dt} = 0$) the ion velocity in the magnetic field coordinate system becomes (see Brekke, 2013; Heinselman and Nicolls, 2008)

$$
v_\mathrm{x} = u_\mathrm{x} + \frac{1}{1 + \kappa_i^2} \left[ \frac{\kappa_i}{B} E_\mathrm{x} - \kappa_i \left( u_\mathrm{y} + \frac{\kappa_i}{B} E_\mathrm{y} \right) - \kappa_i^2 (u_\mathrm{x}) \right] \tag{17a}
$$
$$
v_\mathrm{y} = u_\mathrm{y} + \frac{1}{1 + \kappa_i^2} \left[ \frac{\kappa_i}{B} E_\mathrm{x} + \kappa_i \left( u_\mathrm{x} + \frac{\kappa_i}{B} E_\mathrm{x} \right) - \kappa_i^2 (u_\mathrm{y}) \right] \tag{17b}
$$
$$
v_\mathrm{z} = u_\mathrm{z} + \frac{\kappa_i}{B} E_\mathrm{z}. \tag{17c}
$$

Here, $\kappa_i$ is the ion mobility

$$
\kappa_i = \frac{q_i B}{m_i \nu_{in}}, \tag{18}
$$



the subscript z denotes direction along (antiparallel to) the magnetic field, x horizontally towards east, and y perpendicular to the other two directions, giving a right-handed system. Since we in this article only are considering the ion mobility we will drop the subscript from now on.

The component equations can be combined into a compact matrix equation (see Heinselman and Nicolls, 2008).

$$\boldsymbol{v} = \frac{\kappa_i}{B}\mathbf{C}\boldsymbol{E} + \mathbf{C}\boldsymbol{u}, \tag{19}$$

where $\mathbf{C}$ is the matrix

$$\mathbf{C} = \begin{bmatrix} \frac{1}{1+\kappa^2} & \frac{\kappa}{1+\kappa^2} & 0 \\ \frac{-\kappa}{1+\kappa^2} & \frac{1}{1+\kappa^2} & 0 \\ 0 & 0 & 1 \end{bmatrix}. \tag{20}$$

When estimating the neutral wind and electric field at an altitude Eq. (19) has to be solved. This is an underdetermined inverse problem with six unknowns, which are all components of both the electrical field and the neutral wind velocity. For measurements, we only have the three components of the ion velocity. To resolve this, some a priori assumptions or constraints

are required.

The original solution of Brekke et al. (1973) was to use the fact that $\kappa \gg 1$ at F region altitudes, therefore the ion the ion drift is determined only by the electric field. Then this is assumed to be constant along the magnetic field line. However, the electrical field may not be constant (f.ex. Sangalli et al., 2009). Such an assumption then effects the neutral wind estimates.

It is possible to assume that the neutral wind and electric field vary smoothly in the whole range of interest and use the full

profile of all ion wind measurements to obtain estimates of the neutral wind and electric field. We will outline a procedure to specify a smoothness constraint based on Maxwell's equations in order to give a physically feasible solution.

We start by discretizing the problem as follows: We have a set of ion wind velocity vectors $\boldsymbol{v}_i...\boldsymbol{v}_H$ which are measurements of Eq. (19) integrated over a height range defined by the weighting functions $d_i(h)$

$$\boldsymbol{v}_i = \int_{-\infty}^{\infty} \left[ \frac{\kappa}{B}\mathbf{C}\boldsymbol{E} + \mathbf{C}\boldsymbol{u} \right] d_i(h)\mathrm{d}h + \boldsymbol{\varepsilon}_i, \tag{21}$$

where $\boldsymbol{\varepsilon}_i$ is the noise in measurement $i$ and assumed to be normally distributed with zero mean and covariance described by Eq. (6) (in magnetic field coordinates). We assume that the unknowns can be described by a set of basis functions

$$\boldsymbol{E}(h) = \sum_{j=1}^{N_E} \kappa(h)\beta_j \boldsymbol{b_j}(h)/B(h) \tag{22}$$

and

$$\boldsymbol{u}(h) = \sum_{j=N_E+1}^{N_E+N_u} \beta_j \boldsymbol{b_j}(h). \tag{23}$$

This allows Eq. (21) to be written as

$$\boldsymbol{v}_i = \sum_{j=1}^{N} \boldsymbol{a}_{ij}(h)\beta_j + \boldsymbol{\varepsilon}_i, \tag{24}$$





where

$$\boldsymbol{a}_{ij} = \int\limits_{-\infty}^{\infty} \frac{\kappa(h)\mathbf{C}(h)\boldsymbol{b_j}(h)d_i(h)}{B(h)} dh \tag{25}$$

which can be calculated before solving the problem, and therefore can be regarded as constants.

We assume that the weighting functions $d_i(h)$ for the ion wind measurements are boxcars with centre at a certain height and extending exactly halfway to the centre of the nearest box in both directions. At the ends, the measurement height boxes are symmetric around their centre. The basis functions for the unknowns $\boldsymbol{b}_j(h)$ are also boxcars. Other basis functions could also be used. We further assume that $p$ and the rotation matrix $\mathbf{C}$ are constant throughout our measurement height boxes $d(h)$.

Equation (24) in matrix form then can be written as follows:

$V = \mathbf{A}\boldsymbol{x} + \boldsymbol{\xi},$ \hfill (26)

where $\boldsymbol{V}^\top = [\boldsymbol{v}_1^\top, ..., \boldsymbol{v}_H^\top]$, $\mathbf{A} = \begin{bmatrix} \boldsymbol{a}_{1,1} & \cdots & \boldsymbol{a}_{1,N} \\ \vdots & \ddots & \vdots \\ \boldsymbol{a}_{H,1} & \cdots & \boldsymbol{a}_{H,N} \end{bmatrix}$, $\boldsymbol{x}^\top = [\beta_1, ..., \beta_N]$, and $\boldsymbol{\xi}^\top = [\boldsymbol{\varepsilon}_1^\top, ..., \boldsymbol{\varepsilon}_H^\top]$.

In order to regularize the problem, we use Gauss' and Faraday's laws. Faraday's law for a static magnetic field, $\boldsymbol{\nabla}\boldsymbol{E} = \boldsymbol{0}$, gives us three equations for the gradient of the electric field:

$$\frac{\mathrm{d}E_\mathrm{y}}{\mathrm{d}z} - \frac{\mathrm{d}E_\mathrm{z}}{\mathrm{d}y} = 0 \tag{27a}$$

$$\frac{\mathrm{d}E_\mathrm{x}}{\mathrm{d}z} - \frac{\mathrm{d}E_\mathrm{z}}{\mathrm{d}x} = 0 \tag{27b}$$

$$\frac{\mathrm{d}E_\mathrm{y}}{\mathrm{d}x} - \frac{\mathrm{d}E_\mathrm{x}}{\mathrm{d}y} = 0 \tag{27c}$$

Gauss' law for a charge-neutral plasma, $\boldsymbol{\nabla} \cdot \boldsymbol{E} = 0$, can be written as

$$\frac{\mathrm{d}E_\mathrm{x}}{\mathrm{d}x} + \frac{\mathrm{d}E_\mathrm{y}}{\mathrm{d}y} + \frac{\mathrm{d}E_\mathrm{z}}{\mathrm{d}z} = 0. \tag{28}$$

Equations (27a), (27b) and (28) are added to the theory matrix $\mathbf{A}$ to regularize the electric field. The derivatives $\frac{\mathrm{d}E_x}{\mathrm{d}z}$, $\frac{\mathrm{d}E_y}{\mathrm{d}z}$,

and $\frac{\mathrm{d}E_z}{\mathrm{d}z}$ are approximated with finite differences, with $\mathrm{d}z$ equal to the range step.

The horizontal gradients $(\frac{\mathrm{d}E_x}{\mathrm{d}x} + \frac{\mathrm{d}E_y}{\mathrm{d}y}, \frac{\mathrm{d}E_z}{\mathrm{d}x}, \frac{\mathrm{d}E_z}{\mathrm{d}y})$ are not specified by our measurements. We therefore treat them as Gaussian random variables $\xi_{j,(\mathrm{x,y,z})}$ with zero mean and some variance $\alpha_{j,(\mathrm{x,y,z})}^{-2}$. Eqs. (27a), (27b) and (28) then results in

$$\frac{E_{j,(\mathrm{x,y,z})} - E_{j+1,(\mathrm{x,y,z})}}{\Delta h_E} = \xi_{j,(\mathrm{x,y,z})}. \tag{29}$$

These equations are added to the theory matrix. This implies that we assume these three derivatives of the electrical field

to be smaller than $2/\alpha_{j,(\mathrm{x,y,z})}$ at 95% of the time. For a box size of $\Delta h_E$, this means that $(E_{j,(\mathrm{x,y,z})} - E_{j+1,(\mathrm{x,y,z})}) \sim \mathcal{N}(0, \Delta h_E^2 \alpha_{j,(\mathrm{x,y,z})}^{-2})$, which is similar to first order Tikhonov regularization, but with a regularization constant $\alpha_{j,(\mathrm{x,y,z})}$ that





varies with both height and electric field component. It is worth pointing out that the constraints are obtained from Maxwell's equations and therefore have a physical interpretation.

Constraints, such as Eq. (29), will favour smoother solutions that are closer to constant-valued (Aster et al., 2013). Throughout this paper, we will loosely use "flatness" to describe how close a function is to a constant value, as the magnitude of the left hand side of Eq. (29) is minimized when the function is constant.

For the neutral wind, we also use first order Tikhonov regularization as described for the electric field above. In addition, we use zeroth order Tikhonov regularization to restrict the neutral wind to smaller magnitudes. This corresponds to the following statistical assumptions

$$
\begin{aligned}
u_{j,(\mathrm{x,y,z})} - u_{j+1,(\mathrm{x,y,z})} &= \zeta_{1,j,(\mathrm{x,y,z})} \\
u_{j,(\mathrm{x,y,z})} &= \zeta_{0,j,(\mathrm{x,y,z})}
\end{aligned}
\tag{30}
$$

where $\zeta_{1,j,(\mathrm{x,y,z})} \sim \mathcal{N}(0, \Delta h_u^2 \gamma_{1,j,(\mathrm{x,y,z})}^{-2})$ and $\zeta_{0,j,(\mathrm{x,y,z})} \sim \mathcal{N}(0, \gamma_{0,j,(\mathrm{x,y,z})}^{-2})$. The first row regularizes to the flatness of the profile and the second constrains the magnitude.

This procedure can be interpreted as adding equations for the derivatives of the unknowns to the theory matrix, where these equal to zero with some uncertainty variance justified by physics. This gives us a problem with smooth, well-behaved solutions provided that the constraints are strong enough.

The regularized linear least-squares solution of the inverse problem is then

$$
\hat{\boldsymbol{x}} = \left(\mathbf{A}_R^\top \boldsymbol{\Sigma}_m^{-1} \mathbf{A}_R\right)^{-1} \mathbf{A}_R^\top \boldsymbol{\Sigma}_m^{-1} \boldsymbol{m},
\tag{31}
$$

where $\boldsymbol{m}$ is the extended measurement vector $\boldsymbol{m}^\top = [\boldsymbol{V}^\top \boldsymbol{0}^\top]$ and $\mathbf{A}_R$ is the theory matrix $\mathbf{A}$ extended with the constraints (29) and (30). We will discuss the measurement error covariance matrix $\boldsymbol{\Sigma}_m$ in the next subsection.

## 3.1 Uncertainty calculations

The measurement uncertainty of the ion wind vector estimate at a range $i$ is quantified by the covariance matrix $\boldsymbol{\Sigma}_{v_i} = \left(\mathbf{K}_i^\top \mathbf{K}_i\right)^{-1} \sigma_{w_i}^2$, see Eq. 6. When we combine measurements from different heights to a single vector, the covariance matrix becomes a block matrix with all individual covariances $\boldsymbol{\Sigma}_{v_i}$ along the diagonal,

$$
\boldsymbol{\Sigma}_V = \begin{bmatrix} \boldsymbol{\Sigma}_{v_1} & \cdots & \mathbf{O} \\ \vdots & \ddots & \vdots \\ \mathbf{O} & \cdots & \boldsymbol{\Sigma}_{v_H} \end{bmatrix},
\tag{32}
$$

where $\mathbf{O}$ is the zero matrix. This assumes that measurements from different heights do not correlate.

When expanding the theory matrix to include the regularizations, we also have to expand the covariance matrix. The inverse problem is regularized with a set of values $\alpha_{j,(\mathrm{x,y,z})}$, and $\gamma_{(0,1),j,(\mathrm{x,y,z})}$ which control the smoothness of electric field and neutral wind as a function of height. The values we use for the regularization also form the uncertainty of the added measurements. They are, however, not assumed to be co-varying, and therefore these only add diagonal terms:

$$
\boldsymbol{\Sigma}_L = \mathrm{diag}\left\{\alpha_{1,\mathrm{x}}^{-2}, \alpha_{1,\mathrm{y}}^{-2}, \alpha_{1,\mathrm{z}}^{-2}, \cdots, \alpha_{N,\mathrm{x}}^{-2}, \alpha_{N,\mathrm{y}}^{-2}, \alpha_{N,\mathrm{z}}^{-2}\right\}.
\tag{33}
$$



**Figure 3.** $1\sigma$ uncertainty of estimates of electrical field (a,c,e) and neutral wind (b,d,f). The left column (a and b) shows the perpendicular east components, the middle column shows the perpendicular north components (c and d), and the right column shows the field-aligned components (e and f). The solid lines shows the results for the daytime profile, the dotted lines are for the night profile, and the dashed lines show results for the nighttime profile with aurora. The colours show different regularization parameters. The cyan lines use the numbers derived from Sangalli et al. (2009), the yellow line shows results where the variation in electric field is one tenth, and the blue lines a thousandth of these.





The covariance matrix of the regularized measurements then becomes

$$\mathbf{\Sigma}_m = \begin{bmatrix} \mathbf{\Sigma}_V & \mathbf{O} \\ \mathbf{O} & \mathbf{\Sigma}_L \end{bmatrix}. \tag{34}$$

As the inverse problem then should be solvable using Eq. (31), the uncertainty of the solution is given by

$$\mathbf{\Sigma}_{\hat{x}} = \left( \mathbf{A}_R^\top \mathbf{\Sigma}_m^{-1} \mathbf{A}_R \right)^{-1}. \tag{35}$$

This can be considered as the a posteriori estimation error covariance for the electric field and neutral wind.

## 3.2 Regularization parameters

Before calculating the electric field and neutral wind estimate uncertainties by inserting values into the equation, assumptions must be made on how strongly the problem should be regularized. With ISR, the variation in electrical field and the neutral wind have typically been measured in their own height ranges, neutral wind up to around 140 km, and electrical field above that.

Knowledge on the variation at the other heights is sparse and it is therefore not obvious what good choice for the regularization constants $\alpha$ for the electrical field or $\gamma$ for the neutral wind would be.

Simultaneous observations of electric field and neutral wind have been made with sounding rockets. However, there are not sufficiently many such measurements to fully characterize the statistics of the altitude variation of electrical fields and neutral winds. Altitude profiles of electric field and neutral wind can still be used for estimating typical magnitudes of their gradients in

order to find suitable values for the regularization parameters $\alpha$ and $\gamma$. Here, we will use measurements from the Joule II rocket campaign where altitude profiles from 85 to 210 km of electric field and neutral winds below 130 km were derived (Sangalli et al., 2009). Since the rocket did not travel exactly along the magnetic field line, the variation of the electric fields along the trajectory is larger than along the magnetic field. Therefore, the variance of the electric field gradient will be overestimated, leading to a softer regularization.

At higher altitudes, the electrical field is expected to be constant along the magnetic field because of the high field-aligned conductivity. We therefore use two estimates of the variation of the electrical field, one for high and one for low altitudes. We assume that the variance is the same for the three components. Based on the Sangalli et al. (2009) measurements, we estimate that the largest electric field variation is 20 mV/m over a 2.5 km range at about 90 km and 5 mV/m over the same range at 190 km altitude. We set the regularization parameters to match these variations. This means that we assume that the largest

variations in electric field measured in the rocket experiment are relatively rare (occur 5% of the time). Our regularization of the field-aligned gradient is then $\alpha^{-1}_{j,(x,y,z)} = 1$ μV/m² at 190 km and 4 μV/m² at 90 km altitude. In-between these, we interpolate the variation linearly. We choose our measurement region to be similar, between 80 and 200 km height, and can also extrapolate the variation linearly. This we will call the "measurement-based" regularization. Additionally, we have calculated the uncertainty for two cases where we constrain the electrical field more strongly towards flatness. This can be seen as more similar to the

commonly used assumption that the electrical field is constant. We do this by dividing the regularization for the E-field by 10 and 1000.



We assume a $1\sigma$ variation of the neutral wind of 20 m/s per km for all heights. In addition, we add an assumption that the neutral wind estimates follow a normal distribution with zero mean and standard deviation of 200 m/s, which corresponds to using 0.005 s/m as the zeroth order Tikhonov regularization parameter.

### 3.3 Ion-neutral collision frequency

Use of the right ion-neutral collision frequencies is crucial for calculating the ion mobilities $\kappa_i$ correctly, and therefore it is necessary for estimating the electric field and neutral wind. The collision frequency can be theoretically calculated (see Schunk and Nagy, 2009). It is also possible to measure the collision frequency between ions and uncharged particles $\nu_{in}$ with ISR (Nicolls et al., 2014a). Both of these will result in uncertainty of the collision frequency. In this study, we will ignore this
uncertainty. Any uncertainty in the collision frequency will add to the error budget.

In this study, we have calculated the collision frequencies using

$$\nu_{in} = \frac{\sum_{i,j} s_{i,j} n_i n_j}{\sum_i n_i}, \tag{36}$$

where $s_{i,j}$ is the collision frequency coefficient (CFC) between ion species $i$ and neutral species $j$. We use the CFCs from Schunk and Nagy (2009) for the most usual neutral and ion species $N_2, O_2, O$ and $NO^+, O^+, O_2^+$. Where the collision is
resonant, we simply assume a reduced temperature of 400 K to calculate the CFC. The particle densities are calculated by the MSIS atmospheric model (see Picone et al., 2002).

### 3.4 Electric field and neutral wind uncertainty

We can now investigate the expected performance of E3D for estimating electric fields and neutral winds as a function of height. The variances of the estimates are the diagonal of the a posteriori covariance matrix, Eq. (34). As the performance
depends on ionospheric conditions, we study the same three ionospheric conditions as for the ion velocity uncertainty (see Fig. 1). The performance also depends on the a priori smoothness constraints. Figure 3 shows these $1\sigma$ uncertainties for the different ionospheric conditions and regularization constraints. The ionospheric conditions are indicated with line style. In this figure, the different smoothness assumptions are indicated with color. Cyan is the measurement-based regularization $\alpha$, which is defined in Sect 3.2. We also use two increasingly stronger regularization constraints for the electric field, the yellow line uses
$10\alpha$ and the blue uses $1000\alpha$. This means that yellow and blue lines are assuming the horizontal gradients of the electric field to be a factor of 10 or 1000 smaller in magnitude than the cyan line.

The uncertainties of the perpendicular electric field (Figs. 3a and c) can be divided into two regions: above and below approximately 125 km. Above 125 km, the electric field uncertainty is primarily defined by measurement uncertainty. Below this height, it is primarily constrained by the regularization as ion velocity is less dependent on electric field due to ion
demagnetization. For the parallel E-field (Fig. 3e), the ionospheric conditions play a smaller role.

At low altitudes the uncertainty is above 10 mV/m for the measurement-based regularization. For the higher altitudes, its size depends on the ionospheric conditions, but is around 1 mV/m in the perpendicular directions and approximately a factor of three lower in field-aligned direction. With stronger constraints towards flatness, the uncertainty decreases, but one has to





remember that this comes at the cost of blurring out smaller scale variations. For the lowest range, the yellow and cyan lines
indicate estimates of the electric field with too large uncertainty to be useful. This means that we can not measure electric field
with a useful accuracy below 125 km unless we can make assumptions of horizontal gradients being less than approximately
4 nV/m² (blue line).

The uncertainties of the neutral wind components are shown in Figs. 3b, d and f. The neutral wind is best estimated to
an accuracy of approximately 10 m/s between 90 and 125 km. Below 90 km, the electron density is typically lower, which
increases LOS ion velocity measurement errors. Above 125 km, the ion-neutral collision frequency decreases rapidly, which
makes the ion drift increasingly independent of the neutral wind. At highest altitudes, the uncertainty is merely constrained by
our assumptions on neutral wind amplitude (200 m/s). Best estimates are obtained at around 100 km altitude.

The usable range of neutral wind measurements depends strongly on the prior assumption on the smoothness of the electric
field. The strongest regularization, corresponding to the smallest horizontal electric field gradient assumption, indicated with
the blue line, leads to a neutral wind uncertainty of less than 30 m/s up to 150 km. However, this altitude is greatly reduced
with less strict prior assumptions on the electrical field gradient, see yellow and cyan lines.

It is important to remember that the results and their uncertainties presume that all assumptions of the flatness of the electrical
field or neutral wind profile are true. If our assumptions on the magnitude of the electric field gradients or neutral wind gradients
are too small, the uncertainties presented are overly optimistic.

We can compare our results with earlier measurements of Dahlgren et al. (2011), which used the tristatic EISCAT UHF to
measure the electric field at 220 km altitude under similar conditions as we used for our aurora case. The experiment setup
was similar except for the radar itself. If we look at the time period between 19:28 and 19:36, the horizontal electric field
components had a magnitude of up to 250 mV/m, but mostly around 30 mV/m. Typical standard deviations are tens of mV/m.
With our model and E3D, such electric fields should be measureable with factor of 10 improvement of uncertainty down to
approximately 125 km.

The earlier mentioned Joule II rocket experiment was accompanied with ion velocity measurements at PFISR, which were
used to estimate the neutral wind at the same heights (Heinselman and Nicolls, 2008). Also here, the ionospheric conditions
look most like our aurora example, but the radar pointed into 7 directions to find the different ion wind components. The neutral
wind profiles were integrated over 15 min, and their uncertainties are similar to those in Figs. 3b and d at the highest altitudes,
but somewhat higher further down.

## 4  Simulated measurement

In order to demonstrate how a electric field and neutral wind estimate profile could look like, we have simulated a E3D
measurement and analyzed it. We based the simulations on the Joule II rocket measurement presented in Sangalli et al. (2009).
During the downleg flight, the rocket measured neutral wind at altitudes 90-130 km by tracing chemical releases. The electric
field was measured already from 210 km altitude. Since Sangalli et al. (2009) did not include field-aligned components, we
used a synthetic profile. We used the electric field and neutral wind profiles to simulate E3D ion velocity measurement with



**Figure 4.** Example simulation of electric field and neutral wind estimates. The estimates were calculated from simulations of ion velocity based on measurements of Sangalli et al. (2009). The layout of the figure is as in Fig. 3. The solid lines shows the results for the daytime profile, the dotted lines are for the night profile, and the dashed lines show results for the nighttime profile with aurora. The colours show different regularization parameters. The cyan lines use the numbers derived from Sangalli et al. (2009), the yellow line shows results where the variation in electric field is one tenth, and the blue lines a thousandth of these. The black line shows the values which were used to simulate the ion velocity. We note that the axes on the plots are different.





noise added from Eq. (26). These simulated ion velocity measurements were used to estimate electrical field and neutral wind. By comparing these with the original data set, we can visualize how good the E3D estimates are. The results are shown in Fig. 4. We use the same regularization schemes as for Fig. 3.

The results confirm that the electric field is estimated well above 125 km, as predicted by the uncertainty estimates in Sect. 3.4. Below 125 km the electric field is not estimated well. For all regularization schemes, the behaviour at lower altitudes is similar. The electric field is estimated to be a constant value corresponding approximately to the value at 125 km altitude, indicating that the regularization contributes with all information of the electric field where ions are demagnetized.

The neutral wind is in general better estimated below 120 km altitude, where it is the largest influencer of the ion wind.
Above approximately 125 km, the neutral wind is not well measured in any of the cases. This is not surprising as the neutral wind has little effect on the ion velocity at higher altitudes.

When using the strongest flatness constraints on the electric field, this causes the estimates of the neutral wind to be more fluctuating than the original values (see e.g. Fig. 4b) We believe that the reason is that the model tries to fit the unknowns to the ion wind measurements, but "knows" a priori that the electric field is constant, so all the variation in ion velocity must be
explained by the neutral wind instead of the electric field. If the constraints on the electric field are relaxed, the estimate of all unknowns is closer to the original values.

At the heights where the regularization plays a smaller role, the deviation from the original values seems similar to the predicted uncertainties shown in Fig. 3.

## 5    Discussion

Earlier ISR studies on neutral wind have assumed that the electrical field is exactly constant along the magnetic field – mainly due to the lack of three dimensional ion vector velocity measurements along the whole radar transmit beam. The technique presented in this paper allows us to relax this assumption with a scheme that arises from Maxwell's equations and assumption of horizontal smoothness of electric field. A special case of our regularization scheme is the case where the electric field is approximately constant as a function of height along a magnetic field line. This corresponds to a very strong smoothness
assumption on horizontal gradients of electric field (see blue line in Figs. 3 and 4). The technique presented in this study can be thus seen as a generalization of the commonly used technique for estimating electric field and neutral wind.

Our results indicate that it will be possible to observe an altitude profile of electric field and neutral velocity using E3D. However, it is only possible to reconstruct either the electric field or the neutral wind at any given altitude region. This is ultimately due to the fact that above an altitude of approximately 125 km, the ion drift is to a large extent determined by
electric field, and nearly unaffected by neutral velocity. Similarly, below 125 km, the ion-drift is primarily determined by neutral wind.

For the future measurements, one important question to solve is what regularization parameters should be used. If the constraints are too weak, the problem is underdetermined and the solution noisy. The classical approach is to assume that the electrical field is constant along the magnetic field line. However, the electric field may not always be the constant. Then, as





the example shows, the neutral wind estimates have to compensate to explain the ion drift measurements due to under-resolved electric field variability. Relaxing the assumption of a constant electric field will in these cases improve the results.

Adjusting the regularization constants must be done with caution, since the problem easily becomes underdetermined. Therefore it is important to justify the choices of regularization. For the electrical field, we used regularization parameters which are estimated from in-situ rocket measurements. However, the optimal values of the regularization parameters for general use are still to be found. For the values we used, only constraining the electric field was not enough, and we also constrained the

gradient of the neutral wind in the same way. Here too, the exact values can be discussed. Forcing the neutral wind velocity gradients to be too small causes the estimates of the neutral wind to fit worse to the ion wind at collision-rich heights. This then increases the noise in the electric field here. If the variation is allowed to be too large, the problem is not solvable. In order to allow for higher variations in the neutral wind, but also to use all information we have about it, we added a size constraint of

200 m/s. As can be seen in the uncertainty plots, this restricts the size of the neutral wind components to become smaller.

In future work, the model can, e.g., be improved in one of the following ways. If there somehow exist measurements of neutral wind or electric field, these can be added additionally as constraints. Such measurements could for example be the movement of meteor smoke, polar mesospheric summer echoes, or other measurements of events in the ionosphere that imply size or direction of neutral wind or electric field. An independent measurement of mean neutral wind can often be obtained up

to about 100 km using meteor radars (e.g., Stober et al., 2018).

In this work, we used the same resolution in time for both electric field and neutral wind. The large mass in the neutral atmosphere causes the neutral wind to vary more slowly than the electric field. Nygrén et al. (2011) took advantage of this to use different time resolutions for the different parameters. In the future, it would be an advantage to include this for our model as well.

The technique discussed in this study can be extended further. With the help of phased array technology, E3D will allow fast antenna scanning to be used to measure how ion vector velocity and electron density varies within a volume of space. This type of a measurement may potentially result in improved estimates of electric field and neutral wind, as more physics based regularization can be added. We can use Gauss' and Faraday's laws without the need to treat the horizontal gradients as unknown random variables, as they will be determined by the measurements. We can also introduce constraints that are not

possible for a one-dimensional profile. It will be possible to apply Ampere's law to enforce current continuity. We can also apply Navier-Stokes to enforce that the neutral wind is approximately consistent with anelastic flow. Estimating electric field and neutral wind within a volume is a topic of future work.

*Code availability.* The code is currently only available on request from the corresponding author

*Author contributions.* JV came up of the idea and coded programs for ISR spectrum and geographic calculations. JS carried out the calcula-
tions and prepared the article draft. All participated in developing the technique and the scientific discussions.



*Competing interests.* The authors declare that they have no competing interests

## Acknowledgements

This research has been supported by the Tromsø Science Foundation as part of the project "Radar Science with EISCAT3D".
The publication charges for this article have been funded by a grant from the publication fund of UiT The Arctic University of
Norway. EISCAT is an international association supported by research organisations in China (CRIRP), Finland (SA), Japan
(NIPR and ISEE), Norway (NFR), Sweden (VR), and the United Kingdom (UKRI)






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
