# Peer review of "Observing electric field and neutral wind with EISCAT 3D"

_Annales Geophysicae, 2021_

## Author Response (AR1)

Comments added after the public discussion are written in *italics* and mostly consist of concrete edits to the manuscript, corrections or clarifications.

**Reply on RC1**

We thank the referee for his review and comments. We will answer the comments in the order they appear.

>> This is a comprehensive and well-written effort to quantify the uncertainties in measuring the electric field and neutral winds to be observed by the new EISCAT-3D ionospheric radar, which is not yet operational.

Thank you.

>> I have only a few relatively minor corrections to suggest:

>> Joule heating is a common phenomenon during geomagnetic storm times at high latitudes. From Figure 1, it seems that the authors do not address Joule heating directly. The authors may wish to comment on this, or at least make clear that Joule heating is not a priority for this study.

The phenomenon of Joule heating is indeed outside of the scope of this study. However, the method and framework will be useful for future measurements of Joule heating, as it provides improved measurements of electric field and neutral wind, which are needed in order to calculate Joule heating rates (see e.g., Aikio et.al. 2012, and references therein).

In Figure 1, we are plotting typical background ionospheric parameters which are used to calculate the uncertainty of the ion velocity measurements and the collision frequency between ions and neutrals. The main contributing factor to ion velocity measurement uncertainty is electron density, and we feel that the three cases we are investigating cover most of the typical situations encountered with the radar.

In the revised manuscript, we will add a discussion of how our results would help with measurements of Joule heating. We will also add more discussion about Figure 1.

*Because we inserted a figure further up in the article, the figure mentioned in these comments is now Figure 2.*

*We added some sentences, one about the influence of the temperature on the uncertainty and one on the choice of experiment. We also added some sentences to the discussion.*

>> L191: "the ion" appears twice.

We will correct this.

>> Figure 3: The different lines are hard to see. Please re-plot more clearly.

We have appended the figure with a larger linewidth. It is hopefully easier to see the line now.

*We changed the linethicknesses and -styles in all plots to make the lines easier to see.*

>> Figure 4 caption: The third sentence mentions solid, dotted and dashed lines, which do not appear in the figure. This sentence should be removed.

We will correct this.

>> L411: "fast antenna scanning" should be "fast beam scanning". A phased-array radar does not have a dish which can scan.

This is true. Thank you for pointing out. We will carry out this change.

**Reply on RC2**

The detailed and encouraging comments are warmly appreciated. We will answer the comments in the order they appear.

>> The study describes methodology to estimate uncertainties for the future observations of electric field and neutral wind by EISCAT 3D. Altitude profile variations in these uncertainties are calculated and analyzed in the E and F region ionosphere. This type of analysis is very important for the science community in the preparations for the new facility. However, it seems like some background information is missing. I encourage the authors to consider the following suggestions.

>> * To meaningfully talk about uncertainties in electric field and neutral wind measurements the actual values of those parameters or their ranges need to be outlined to communicate if the concluded uncertainties are of the order of 1, 10 or 100% of the absolute value.

The uncertainties in the paper are absolute, not relative. This means that in order to draw conclusions, one has to compare with the usual approximate magnitude of electric field and neutral wind. We believed that we had done this in the discussion section, but we interpret the comment as that the discussion is too short.

We will extend the discussion on the magnitude of the uncertainties in the revised manuscript.

>> * The estimated uncertainties are for a time and height resolution of 5 sec and 2 km. How are these selected and how sensitive the results are to the data resolution? E3D is meant to be a versatile instrument, so the resolutions are hardly fixed values there.

This is entirely true, the 5 s integration-time chosen here is purely chosen to be representative of a short integration-period for an alternating-coded long-pulse experiment, how much shorter integration-periods could be used with E3D remains to be seen, likewise the 2 km range-resolution could very well be reduced at F-region altitudes where the field-aligned potential difference should be very small - which would reduce the uncertainty.

In the E region, one would like to have the resolution as best as possible, while this is not necessary for the F region. However, since signal from the F region is weaker, more measurements are needed to give meaningful results.

We therefore see the 2 km range resolution as a compromise for an experiment investigating both E and F region. For simplicity, we kept the range resolution constant.

*We added two sentences similar to the paragraph above*

>> * A short paragraph at the end of the introduction stating the motivation and aim of this study, and the knowledge gap that is being filled with this study would be very helpful.

We will add a such paragraph at the end of the introduction.

>> * Providing a sketch of the scattering geometry together with the station locations would help a non-Scandinavian reader very much.

We agree and will add a figure showing the locations of the E3D sites.

>> * Proper definitions of the "quiet night-time" and "night-time with auroral precipitation" need to be given to provide the context and background for the comparison.

With quiet night-time, we mean a night where there is no auroral precipitation. Here, we simply use the IRI2016 model which does not include auroral precipitation or its effects. We will try to clear up the sentence.

*We changed this part to «night time without aurora as modelled by IRI»*

>> Additional small items:

>> Line 6: either field-aligned profile or altitude profile

We will use field-aligned profile

>> Line 14: How is significant influence determined?
We will change the expression to «dominate» to clear up the sentence.

>> Line 126: It is not obvious what this downscaling means...

The sentence: "This takes into account the recent downscaling of the core array from 109 to about 35 subarrays for the first stage of E3D" unfortunately is not explained sufficiently well.

The earlier plans for EISCAT 3D indicated that there would be 109 antenna modules with transmitters, which we have previously used for radar performance parameter calculations. Due to funding related issues, the most recent EISCAT 3D design reduces the number of antenna modules with transmitters to 35. Due to this reason, the transmit antenna directivity is now estimated to be 38 dB instead of the previous much higher number.

In order to improve the quality of the manuscript text, we will change to "These are to best of our knowledge the performance parameters of the latest revision of the EISCAT 3D design, which may still change before the final implementation".

>> Line 147: Can you put a factor for the lower electron density during the solar minimum conditions?

According to Brekke (2013), the factor is somewhat below 2. We will add this to the revised manuscript.

>> Line 148: Is integrating over a longer time period not another option?

The integration-time is hard-limited by the natural time-variations of the ionospheric variations. At auroral latitudes this is often at time-scales of a couple of seconds to a few tens of seconds. During quiet or stable conditions longer integration-times are a possible option.

When extending the one-beam approach to multiple beams in different directions to find the three-dimensional distributions of winds and fields, it is desirable to have as short integration time as possible.

>> Figure 2: The nighttime profiles seem to have semi-regular wiggles in them. Where do those come from?

We will investigate the source of the wiggles. The plasma parameters for the aurora case are based on noisy EISCAT measurements, which have estimation errors, but the day and night case are based on the IRI profiles, which means that they should be as smooth as the IRI profiles.

>> Line 181: Is dropping the subscripts referring to the xyz ones or the i ones or both?
We are dropping the subscript i from the ion mobility. We will state this clearer.

>> Line 187: "at a certain altitude" rather than "at an altitude"
This will be corrected.

>> Line 188 (and on many other lines): "electric" instead of "electrical"
We will fix this for the revised manuscript

>> Figure 3: The dotted and dashed lines are very difficult to differentiate. Please use different marking/symbols. "Lines show" instead of "line shows" in the last caption sentence.
We hope dashed and dash-dotted lines will clear it.

*All figures are replotted with thicker lines and solid, dashed and dash-dotted lines*

>> Line 285: "by" rather than "in" the rocket experiment
We will fix this for the revised manuscript

>> Line 299: Please state the uncertainties of the previous studies for comparison.
We will do this. From Nicolls et al. (2014), it seems that both ways of estimating the collision frequency has an uncertainty in the size order of 50 %.

**Other possibly important changes**

-In the preprint, the rotation matrix in Eq. (4) was used directly. However, it should have been transposed, as the needed transform is from ion velocity $v$ in local geomagnetic coordinates to geographical coordinates for theory matrix $\mathbf{K}$ and measurements $w$. For clarification, the transform matrices are now also included in the following equations.

The results are mainly unchanged because for calculating the uncertainties, the rotation is taken both ways. Only for Fig. 5 becomes somewhat different. The change is in such a way that it does not change results or conclusion.

-The added figure with the first E3D sites also contains an overview on how the experiment can be carried out. One purpose is to show that the receivers can form several beams at the same time to measure all ranges at once. This is new to E3D and is that what enables the short time resolution in the ion velocity measurements.

- In beginning of Sect. 3, some subscripts were changed to upright characters

- Equation (25), the difference between neutral wind and electric field rows of the theory matrix is added.